# Facile Synthesis of GNPs@Ni_x_S_y_@MoS_2_ Composites with Hierarchical Structures for Microwave Absorption

**DOI:** 10.3390/nano9101403

**Published:** 2019-10-02

**Authors:** Wenfeng Zhu, Li Zhang, Weidong Zhang, Fan Zhang, Zhao Li, Qing Zhu, Shuhua Qi

**Affiliations:** Department of Applied Chemistry, School of Natural and Applied Sciences, Northwestern Polytechnical University, Xi’an 710072, China; zwenfeng89@mail.nwpu.edu.cn (W.Z.); zhangli_nwpu@outlook.com (L.Z.); jofun@mail.nwpu.edu.cn (F.Z.); lizhao1314@mail.nwpu.edu.cn (Z.L.); zhuqing@mail.nwpu.edu.cn (Q.Z.)

**Keywords:** Microwave absorption properties, impedance matching, synergy effect, dielectric loss

## Abstract

Graphene-based powder absorbers have been used to attain excellent microwave absorption. However, it is not clear if inferior microwave absorption by pure graphene materials can be attributed to impedance mismatching or inadequate attenuation capability. In this comparative study, we focus on these aspects. Graphene nanoplatelets (GNPs) multi-component composites (GNPs@Ni_x_S_y_@MoS_2_) were prepared by hydrothermal reaction with different S and Mo molar ratios. The morphologies, phase crystals, elemental composition, and magnetic properties of the composites were also analyzed. In addition, microwave absorption of the as-prepared samples was investigated and it revealed that the impedance mismatching could be responsible for inferior microwave absorption; higher conductivity can lead to skin effect that inhibits the further incidence of microwaves into the absorbers. Furthermore, the optimum reflection loss (RL) of GNPs@Ni_x_S_y_@MoS_2_-2 can reach −43.3 dB at a thickness of 2.2 mm and the corresponding bandwidth with effective attenuation (RL < −10 dB) of up to 3.6 GHz (from 7.0 to 10.6 GHz). Compared with the GNPs, the enhanced microwave absorption can be assigned to the synergistic effects of conductive and dielectric losses.

## 1. Introduction

Electromagnetic (EM) pollution by the modern industry has become inevitable because EM radiation harms health, military devices performance, and information transmission. To alleviate this harm, various EM absorbing materials have been developed. Amongst them, magnetic ferrites [1] were the first to be used, followed by linear carbon-based materials such as carbon fibers [2], carbon nanotubes [3], and conductive polymers [4,5,6]. However, absorbers developed using these traditional materials have a narrow absorbing bandwidth. New absorber materials with high microwave absorption performance (MAP) are therefore necessary.

The absorber’s structure is one of the factors that affects the EM wave attenuation [7,8]. Materials with hierarchical structures are likely to possess a good MAP. Hence, materials such as graphene [9,10,11,12], MoS_2_ [7,13,14,15,16,17], and MXenes [18,19,20,21] are selected as EM absorbers. Heterostructure MoS_2_ hybrids such as MoS_2_/Ni nanoparticles [22], MoS_2_/carbon nanotubes [23], core-shell NiS_2_@MoS_2_ nanosphere [24], and self-assembled carbon sphere coated by MoS_2_ nanosheets (CS/MoS_2_) [13] are facile candidates for fabricating absorbers to reduce EM pollution. This is attributed to their sheet-like morphology, which can generate larger interfacial polarization and high dielectric loss. To date, MoS_2_ composites are still effective high-performance EM absorbers [16].

Impedance matching is presumably another key factor for materials’ MAP; this explains why semiconductor materials are preferred over insulators for high MAP absorbers. Metal sulfides and their composites are associated with excellent EM absorption due to their higher dielectric loss than metal oxides [25,26]. Hierarchical hollow CuS nanoparticles, according to Zhao and co-workers’ report, can be good absorbers with a minimum reflection loss (RL) value of −17.5 dB and effective bandwidth of 3.0 GHz [27]. Core-shell nanostructured NiS_2_@MoS_2_, physically combined by NiS_2_ and MoS_2_, can achieve an excellent microwave absorption ability of −41.05 dB at 12.08 GHz with a thickness of 2.2 mm [24]. Ni@Ni_2_S_3_ foam reportedly has a wide range of effective attenuation bandwidth (12.75–18.0 GHz) and an intense reflection loss (−50.7 dB) with a thickness of 3.6 mm [28]. Interestingly, metal sulfide composites have better MAP than a single metal sulfide component due to its lower extent of impedance matching [29].

Considering nano-structural engineering and impedance matching [30,31,32,33,34,35,36], we present novel and high-efficiency absorbing metal sulfides on graphene nanoplatelets (GNPs@Ni_x_S_y_@MoS_2_) with hierarchical structures. Firstly, we deposited nickel particles on GNPs through a simple electroless nickel-plating method. Through the sulfurization process, we used GNPs@Ni, sodium molybdate, and thioacetamide as the precursors for nickel, molybdenum, and sulfur source, respectively, to obtain the GNPs@Ni_x_S_y_@MoS_2_ composites with hierarchical structures from a single-step procedure of facile hydrothermal reaction. For ideal absorbers, different hierarchical structures were obtained by controlling S/Mo molar ratios. Studies on morphologies and EM absorption properties suggest that the as-prepared GNPs@Ni_x_S_y_@MoS_2_ with hierarchical nanostructures have optimized morphology and impedance matching, strong dielectric loss, and interface polarization, thus resulting in excellent MAP.

## 2. Experimental

### 2.1. Materials

Sodium molybdate (Na_2_MoO_4_·2H_2_O, AR) and thioacetamide (C_2_H_5_NS) were bought from Janus New-Materials Co., Ltd. (Nanjing, China). Oxidized graphite (OG) was supplied by Qingdao Yanxin Graphite Products Co., Ltd. (Qingdao, China). Sodium hydroxide (NaOH, AR), stannous chloride (SnCl_2_, AR), hydrochloric acid (HCl, 36%, AR), palladium dichloride (PdCl_2_, AR), sodium hypophosphite (NaH_2_PO_2_·H_2_O, AR), nickel sulphate hexahydrate (NiSO_4_·6H_2_O, AR), ammonia solution (NH_3_·H_2_O, 28%, AR), and malic acid (C_4_H_6_O_5_, AR) were all obtained from Sinopharm Chemical Reagent Co., Ltd. (Shanghai, China). Ethanol (C_2_H_5_OH, AR) was provided by Fuyu Fine Chemical Reagent Co., Ltd. (Tianjin, China). All reagents were used without further purification.

### 2.2. Synthesis of GNPs@Ni Composite

Graphene nanoplatelets (GNPs) were prepared as in a former report [37] and GNPs@Ni was obtained in four steps as follows: Firstly, 0.5 g of GNPs was added into 100 mL 2 M NaOH aqueous solution at 45 °C for 2–5 h to clean the surface of GNPs and to introduce hydrophilic groups [38]. Secondly, the GNPs were sensitized by 50 mL SnCl_2_/HCl mixed solution with the concentrate 1 wt% and containing 1 wt% HCl, then continuous stirring for 1 h [39]. Thirdly, the sensitized GNPs were immersed in 50 mL PdCl_2_·2H_2_O (0.5 g/L)/HCl (10 g/L) mixed solution for another hour at room temperature (RT). Finally, the as-treated GNPs were transferred into a Ni-plating solution containing nickel sulphate (25 g/L), sodium hypophosphite (30 g/L), malic acid (30 g/L), and an excess NH_3_·H_2_O (28%). The Ni-plating reaction was kept at 60 °C for 1 h. The products were collected and a robust yield calculated by mass showed that the nickel mass ratio (mass of nickel in GNPs@Ni composite) was approximately 38%.

For comparison, the Ni nanoparticles were prepared by mixing the solution of nickel sulphate (25 g/L), sodium hypophosphite (30 g/L), malic acid (30 g/L), and an excess NH_3_·H_2_O at 60 °C for 1 h.

### 2.3. Synthesis of GNPs@Ni_x_S_y_@MoS_2_ Composites

In a typical procedure, 0.1 g of as-prepared GNPs@Ni, 0.242 g of sodium molybdate (Na_2_MoO_4_ ·2H_2_O, 1 mmol), and a certain amount of thioacetamide (C_2_H_5_NS) were dissolved in 120 mL of deionized water under sonication for 30 min. The resulting solution was then mixed and transferred into a 200 mL Teflon-lined stainless-steel autoclave and kept at a temperature of 220 °C for 10 h. It was then cooled to room temperature. After the products were collected by centrifugation they were washed with pure water and ethanol several times. Freeze-drying resulted in GNPs@Ni_x_S_y_@MoS_2_ composites with three different amounts of C_2_H_5_NS (4 mmol, 6 mmol, and 8 mmol, designated as GNPs@Ni_x_S_y_@MoS_2_-1, GNPs@Ni_x_S_y_@MoS_2_-2, and GNPs@Ni_x_S_y_@MoS_2_-3 with the corresponding S/Mo ratios as 4, 6, and 8, respectively).

For comparison, GNPs@Ni_x_S_y_ composites were also prepared under the same experimental conditions with GNPs@Ni_x_S_y_@MoS_2_-1; however, Na_2_MoO_4_ ·2H_2_O was excluded this time. According to the experiment record, the weight increasing of GNPs@Ni_x_S_y_ composites, GNPs@Ni_x_S_y_@MoS_2_-1, GNPs@Ni_x_S_y_@MoS_2_-2, and GNPs@Ni_x_S_y_@MoS_2_-3 were 20, 170, 175, and 174%, respectively. Thus, based on the simple calculation, we can infer the composition of as-prepared samples, asshown in Appendix A.

### 2.4. Characterization

The morphologies of the as-prepared samples (GNPs@Ni and GNPs@Ni_x_S_y_@MoS_2_ composites) were displayed by the FE-SEM Hitachi S4800 microscope (Hitachi, Ltd., Tokyo, Japan). The TEM images of GNPs@Ni_x_S_y_@MoS_2_-3 were carried out by the JEOL JSM-2010 microscope (JEOL Ltd., Tokyo, Japan). Meanwhile, crystal structures of GNPs@Ni_x_S_y_ and GNPs@Ni_x_S_y_@MoS_2_ composites were analyzed by the Bruker D8 Advanced X-ray (Bruker Corporation, Karlsruhe, Germany) in the range of 2θ = 5–80°. On the other hand, elementary compositions of the composites were surveyed by an XPS spectra with the PHI 5000 Versa Probe (ULVAC-PHI, Inc. Chigasaki, Japan). The electrical conductivities (*σ*) of all the samples were obtained with RTS-8 (4-Probes-Tech, Guangzhou, China). Thermogravimetric analysis (TGA) was performed on Netzsch STA-449F3 Thermal Analyzer (NETZSCH Companies, Selb, Germany) at air atmosphere with the heating rate of 10 °C min^−1^ in air atmosphere. The magnetic property was tested with a physical property measurement system (CFMS-14T, Nikkiso Cryogenic Industries Group, Temecula, CA, USA). The research on EM parameters were carried out with a vector network analyzer (Agilent PNA N5224A, Agilent Technologies, Santa Clara, CA, USA) using the coaxial wire method within the range of 2.0–18 GHz. The samples used for EM parameter measurement were prepared by mixing the GNPs@Ni_x_S_y_@MoS_2_ composites with paraffin with a mass ration of 70%. The mixtures were then cold-pressed into a toroid (Φ_out_ = 7.0 mm, Φ_in_ = 3.04 mm) at a proper pressure.

## 3. Results and Discussion

### 3.1. XRD Analysis

Phase crystals investigation of the products is performed by XRD. Figure 1a shows the as-obtained XRD pattern of GNPs@Ni_x_S_y_ composites and the pattern shows two phases in the sample. The diffraction peaks located at 2θ = 16.2° (111), 26.6° (220), 31.3° (311), 37.9° (400), 46.9° (422), 49.9° (511), 54.7° (440), 64.4° (533), 68.5° (444), 74.9° (642), 77.3° (731) are assigned to polydymite Ni_3_S_4_ (JCPDS Card no. 47–1739). Other peaks at 2θ = 35.3° (210), 38.8° (211), 45.3° (220), 53.6° (311), 56.3° (222), 58.8° (230), 61.2° (321) can be perfectly indexed to the vaesite NiS_2_ (JCPDS Card no. 11–0099). After the sulfurization process, the three main diffraction peaks of Ni disappeared at 2θ = 44.5°, 51.8°, and 76.4°. NiS_2_ and Ni_3_S_4_ peaks, however, appeared, thus indicating that metallic Ni was completely transformed into NiS_2_ and Ni_3_S_4_ composites. The strong and sharp diffraction peaks confirmed that the obtained products were properly crystallized [29].

For GNPs@Ni_x_S_y_@MoS_2_ composites (Figure 1b), after excluding the diffraction pattern of NiS_2_ and Ni_3_S_4,_ the broad diffraction peaks at 14.4°, 33.5°, and 39.5° are attributed to the diffraction pattern of the MoS_2_ (002), (101), (103), respectively, (JCPDS Card no. 37–1492) [8]. Thus, the dominating composites of Ni_x_S_y_ are polydymite Ni_3_S_4_ and vaesite NiS_2_ during the hydrothermal reaction and the existence of Ni_3_S_4_, NiS_2_, and MoS_2_ was confirmed in GNPs@Ni_x_S_y_@MoS_2_ composites.

### 3.2. XPS Analysis

As shown in Figure 2, the oxidation states of elements and the surface composition of the GNPs@Ni_x_S_y_@MoS_2_ composites were further confirmed by XPS. Figure 2a displays the survey spectra of GNPs@Ni_x_S_y_ and GNPs@Ni_x_S_y_@MoS_2_. As expected, Mo, S, C, and Ni signals were observed in the survey spectra. The peaks at approximately 227–240 eV and 400–410 eV can be ascribed to Mo 3d and Mo 3p. Other 2 peaks at 229.2 and 232.5 eV in Figure 2b aligns with Mo 3d_5/2_ and Mo 3d_3/2_ of MoS_2_. The peak located at 226.3 eV corresponds to S 2s. In the spectrum of S 2p (in Figure 2c), the peaks at 161.7 and 162.5 eV are attributed to the S 2p_3/2_ and S 2p_1/2_. Core level spectra of Mo and S, shown in Figure 2b,c, confirm the presence of MoS_2_ [40,41,42]. The binding energies of 855.4 and 874.4 eV corresponding to Ni 2p_3/2_ and Ni 2p_1/2_ are caused by charge-transfer screening and attributed to the presence of Ni cations from Ni_3_S_4_ and NiS_2_ (Figure 2d) [24,28,43,44,45,46]. All XPS analyses pointed at the formation of GNPs@Ni_x_S_y_@MoS_2_ composites.

### 3.3. SEM Analysis

The morphologies and structures of the as-prepared GNPs@Ni and GNPs@Ni_x_S_y_@MoS_2_ composites were studied by FE-SEM (Figure 3a–h). As illustrated in Figure 3a,b, numerous nickel particles with nearly spherical shape stacked on the surface of GNPs to form wrinkled skin. The size of Ni particles was uniform with an average diameter of approximately 150–200 nm (see in Figure 3b). Figure 3c shows that GNPs@Ni_x_S_y_@MoS_2_-1 maintained a nearly spherical shape, which is similar to the appearance of GNPs@Ni composites. Meanwhile, Figure 3d displays a clear view of the surface morphology, which reveals that the flower-like Ni_x_S_y_@MoS_2_-1 nanospheres are composed of numerous intercrossed curved nanosheets with a thickness of several nanometers. Due to the laminar growth habit of Ni_x_S_y_@MoS_2_, the agglomerated nuclei tend to self-assemble into a sphere-like microstructure to reduce the interfacial energy in nanosheets [17,47,48,49,50,51]. The wrinkled nanoplates would increase the specific surface area where it is beneficial to obtain higher microwave absorption performance [17,52,53]. When the ratio of S/Mo reached up to 6 (GNPs@Ni_x_S_y_@MoS_2_-2), the as-synthesized Ni_x_S_y_@MoS_2_ nanosheets increased in size to such an extent that the space was insufficient for the nanosheets to assemble into nanospheres (Figure 3e,f). Once the ratio of S/Mo reached 8, a hierarchical structure was observed (see GNPs@Ni_x_S_y_@MoS_2_-3 in Figure 3g,h). This is mainly because the Ni_x_S_y_@MoS_2_-3 nanosheets stacked together and grew anisotropically on the surface of GNPs.

### 3.4. TEM Analysis

The microstructures of the samples were further investigated via TEM analyses. As seen in Figure 4a, nearly sphere-like nickel particles were distributed on the surface of GNPs, thus mimicking the FE-SEM images above. In the case of the GNPs@Ni_x_S_y_@MoS_2_-3 nanocomposite, the TEM image exhibits the presence of Ni_x_S_y_@MoS_2_ nanosheets attached to the surface of GNPs (Figure 4b). The selected area diffraction (SAED) pattern in Figure 4c further confirms the existence of the single crystallinity of nickel sulfide (Ni_x_S_y_) and the (002) planes of polycrystallinity of MoS_2_ in the sample. To search the elementary compositions of GNPs@Ni_x_S_y_@MoS_2_-3, elemental mappings of GNPs@Ni_x_S_y_@MoS_2_-3 are displayed in Figure 4d. It can be seen that C, Mo, and S mappings are evenly distributed. They surround the GNPs@Ni_x_S_y_@MoS_2_-3 frame except for Ni whose mapping is distributed in a corner. This indicates that the Ni nanoparticles only existed in that corner of the GNPs region. These results coincide with the analysis of XRD and XPS patterns.

All the morphological, crystalline, and elementary characterizations above have demonstrated that the hierarchical GNPs@Ni_x_S_y_@MoS_2_ composites were successfully synthesized. Furthermore, different hierarchical structures could be obtained by controlling the ratios of S/Mo, which may be significantly correlated to the MAP.

### 3.5. Magnetic Properties

As presented in Figure 5, the magnetic hysteresis loops of the as-prepared GNPs@Ni composites were S-shaped, with the low coercivity and remanence magnetization indicating that the GNPs@Ni composites are of typical soft magnetic behavior [51,54]. The saturation magnetization (*M_s_*) of GNPs@Ni composite is 13 emu/g, which is lower than that of pure Ni nanoparticles (35 emu/g). The decrease in Ms values is mainly attributed to the demagnetizing field caused by GNPs. According to Equation (1), the weight percentage of Ni over GNPs@Ni is 37.14%, which corresponds to the calculated values obtained from the experiment.

Coating magnetic nanoparticles is one of the effective ways to enhance the material’s MAP because of the role of magnetic loss. Normally, the magnetic loss can be evaluated by initial permeability (*μ_i_*). According to the Equation (2), *μ_i_* values of GNPs@Ni are higher than those of pure Ni. Thus, we can conclude that GNPs with Ni particles can enhance that magnetic loss more than pure Ni.
(1)$$M_{s}=\varphi m_{s}$$
(2)$$\mu_{i}=\frac{M_{S}^{2}}{akH_{c}M_{s}+b\lambda \xi }$$

In the equations, *M_s_* represents saturation magnetization, *H_c_* reflects the maximum coercivity, *a* and *b* are two constants determined by the material composition, *λ* is the magnetostriction constant, and *ξ* is an elastic strain parameter of the crystal [8,55].

### 3.6. Thermogravimetric Analysis (TGA)

Concerning the working environment of the absorber, TGA was performed on Netzsch STA-449F3 Thermal Analyzer at air atmosphere with the heating rate of 10 °C min^−1^ to evaluate the thermal stability of the samples. TG results of GNPs@Ni_x_S_y_@MoS_2_ composites are presented in Figure 6. It can be observed that all the TG curves can be divided into three stages. The first weight loss below 160 °C, which can be assigned to the evaporation of physiosorbed and chemisorbed water. Obviously, the percentage of water is too high, due to the freeze-drying technique without further drying at higher temperature. The second weight loss from 160 °C to 375 °C is mainly due to the thermal decomposition of MoS_2_ and oxidation of Ni_x_S_y_. The third weight loss can be responsible for the complete oxidation of MoS_2_ and GNPs [56,57]. However, it is very interesting that the residual weight is different because of the ratio of S/Mo. Compared with GNPs@Ni_x_S_y_@MoS_2_-1 and -2, we can infer that MoS_2_ is more stable with the increasing of S/Mo. The curves of GNPs@Ni_x_S_y_@MoS_2_-2 and -3 are nearly the same, while the ratio of S/Mo is up to 6 and 8, respectively.

### 3.7. Microwave Absorption Properties

The MAP can be presented by RL value according to transmission line theory, and RL value can be calculated from the corresponding EM parameters, see Equations (3)−(5).
(3)$$Z_{in}=Z_{0}\left(\sqrt{\frac{\mu_{r}}{\varepsilon_{r}}}\right)tanh\left[j\left(\frac{2\pi fd}{c}\right)\left(\sqrt{\mu_{r}\varepsilon_{r}}\right)\right]$$
(4)$$\Gamma =\frac{Z_{in}-Z_{0}}{Z_{in}+Z_{0}}$$
(5)RL=20lg|Γ|
where *Z_0_* is the impedance of air, *Z_in_* is the input impedance of the absorber, *c* is the light velocity, *f* is the frequency of the EM wave, *d* is the thickness of the absorber, and *Γ* is the reflection coefficient of the material [8,58].

The absorber with RL values lower than −10 dB is regarded for practical application [16,37]. The RL calculation results and the corresponding EM parameters of GNPs and GNPs@Ni composites are summarized in Appendix A. Owing to the higher conductivity of Ni nanoparticles, the ε′ values of GNPs@Ni were much higher than those of GNPs (partial of the ε″ values are negative) [59]. The corresponding RL was calculated and plotted in Appendix A, from which we can see that the minimum RL of GNPs@Ni was only −4.5 dB and much lower than that of GNPs (the minimal RL was −25.7dB). The GNPs@Ni_x_S_y_@MoS_2_ composites display excellent microwave absorption abilities (in Figure 7). It is noted that GNPs@Ni_x_S_y_@MoS_2_-1 (Figure 7a,b) has a RL low to −27.1 dB when the absorber thickness is 2.3 mm, and very wide bandwidth with effective attenuation is discovered in the frequency range of 5.8–7.0 GHz. As the ratios of S/Mo increases, the MAP of GNPs@Ni_x_S_y_@MoS_2_ composites is improved (in Figure 7c,d). Figure 7c suggests that the minimum RL of GNPs@Ni_x_S_y_@MoS_2_-2 can reach −43.3 dB at a thickness of 2.2 mm. Simultaneously, the bandwidth with effective attenuation increased to 3.6 GHz (from 7.0 to 10.6 GHz). The 3D surface plots (in Figure 7d) indicate that the effective absorption mainly focuses on 5.0–11.0 GHz with a thickness of 2.0–3.0 mm. However, for GNPs@Ni_x_S_y_@MoS_2_-3 (in Figure 7e,f), the conspicuous minimum RL values of −39.5 dB and −28.3 dB are obtained at the matching thicknesses of 2.4 mm and 4 mm, respectively, and the corresponding effective attenuation bandwidth is observed in the 7.3–9.1 GHz and 4.1–5.2 GHz range. Hence, it is evident that GNPs@Ni_x_S_y_@MoS_2_-2 composites display the best MAP in terms of both the minimum RL value and the effective attenuation bandwidth.

Figure 8 and Appendix A show the EM parameter and corresponding calculation. GNPs@Ni composites show a higher relative complex permittivity than the GNPs@Ni_x_S_y_@MoS_2_ composites with the same filler loading. This indicates that the sulfurization method has a profound influence on the EM parameters [29]. However, the excessive real part (ε′) and imaginary part (ε″) of complex permittivity might influence the impedance matching adversely [24,29,60]. Compared with Figure 8a,b, the ε′ and ε″ of GNPs@Ni_x_S_y_@MoS_2_-1 are higher than those of the other GNPs@Ni_x_S_y_@MoS_2_ samples. Meanwhile, the different hierarchical structures of composites influence the EM parameters as well [16,61]. According to classical EM theory, ε′ and ε″ can be expressed as Equations (6)–(8):(6)$$\varepsilon_{r}=\varepsilon_{\infty }+\frac{\varepsilon_{s}-\varepsilon_{\infty }}{1+j\omega \tau }=\varepsilon^{\prime }-j\varepsilon^{\prime \prime }$$
(7)$$\varepsilon^{\prime }=\varepsilon_{\infty }+\frac{\varepsilon_{s}-\varepsilon_{\infty }}{1+\omega^{2}\tau^{2}}$$
(8)$$\varepsilon^{\prime \prime }=\frac{\varepsilon_{s}-\varepsilon_{\infty }}{1+\omega^{2}\tau^{2}}\omega \tau +\frac{\sigma }{\omega \varepsilon_{0}}=\varepsilon_{p}^{\prime \prime }+\varepsilon_{c}^{\prime \prime }$$
where $\varepsilon_{s}$ and $\varepsilon_{\infty }$ denote the static permittivity and the high-frequency limit permittivity, respectively. σ is the conductivity of the composite, $\varepsilon_{0}$ = 8.854 × 10^−12^ F/m. $\varepsilon_{p}^{\prime \prime }$ and $\varepsilon_{c}^{\prime \prime }$ represent polarization loss and conductivity loss, respectively [8]. 

According to Equation (8), dielectric loss behaviors depend on $\varepsilon_{p}^{\prime \prime }$ and $\varepsilon_{c}^{\prime \prime }$. The $\varepsilon_{c}^{\prime \prime }$ is closely related to *σ*. The high $\varepsilon_{c}^{\prime \prime }$ value represents low electrical resistivity, but higher conductivity always contributes to unsatisfactory microwave absorption abilities because of impedance mismatching. By adjusting the EM parameters of GNPs@Ni through sulfurization, the values of the ε′ and ε″ decline significantly.

Dielectric loss is the unique pathway to attenuate EM wave in GNPs@Ni_x_S_y_@MoS_2_ composites. As a result, conductivity loss, dipole orientation polarization, and interfacial polarization are the possible candidate mechanisms to attenuate EM waves [62,63,64].

The electrical conductivities (*σ*) of the composite samples are listed in Appendix A to research the possible mechanisms of the MAP of the samples. As shown in Figure 8b and Appendix A, the *σ* values for GNPs@Ni_x_S_y_@MoS_2_-1, GNPs@Ni_x_S_y_@MoS_2_-2, and GNPs@Ni_x_S_y_@MoS_2_-3 are 3.3433 S/cm, 1.7177 S/cm, and 0.5258 S/cm. The higher ε″ of GNPs@Ni_x_S_y_@MoS_2_-1 nanocomposite mainly originates from the higher $\varepsilon_{c}^{\prime \prime }$, indicating that conductive loss is a prominent mechanism to attenuate EM wave [65].

In addition, the important parameters of dielectric loss tangent (tanδ_ε_ = ε″/ε′) and attenuation constant (α) are displayed in Figure 8c,d and Appendix A. The value of the α is given by
(9)$$\alpha =\frac{\sqrt{2}\pi f}{c}\times \sqrt{\left(\mu^{\prime \prime }\varepsilon^{\prime \prime }-\mu^{\prime }\varepsilon^{\prime }\right)+\sqrt{{\left(\mu^{\prime \prime }\varepsilon^{\prime \prime }-\mu^{\prime }\varepsilon^{\prime }\right)}^{2}+{\left(\mu^{\prime }\varepsilon^{\prime \prime }+\mu^{\prime \prime }\varepsilon^{\prime }\right)}^{2}}}$$

GNPs@Ni shows the highest values of tanδ_ε_ and α, which are in good agreement with the high values of *ε″*. Besides, as displayed in Figure 8c,d, both tanδ_ε_ and α of GNPs@Ni_x_S_y_@MoS_2_-1 are higher than those of other GNPs@Ni_x_S_y_@MoS_2_ samples and the MAP of GNPs@Ni_x_S_y_@MoS_2_ composites are conflicting with the corresponding conductivity because the impedance matching property was ignored [61].

Concerning impedance matching properties ($Z=\left|Z_{\mathrm{in}}/Z_{0}\right|$), the ideal situation is *Z* = 1. According to the following calculation,
$$\mathrm{RL}=20\mathrm{lg}\lfloor \frac{Z_{in}-Z_{0}}{Z_{in}+Z_{0}}\rfloor \leq -10\mathrm{dB}$$
1≥ZinZ0≥10−110+1≈0.52
if the attenuation property is not the limiting factor for EM wave absorption, the effective attenuation occurs between 0.52 ≤ *Z* ≤ 1 (denotes as impedance matching area). The contour maps of *Z* and RL of GNPs and as-prepared GNPs-based composites are shown in Figure 9 and Appendix A. GNPs show inferior impedance matching because of the tiny impedance matching area (Appendix A). After incorporating Ni nanoparticles on the surface of GNPs, GNPs@Ni composites showed the distinct impedance mismatching due to the impedance matching area (Appendix A), which is a good explanation for the poor MAP of GNPs@Ni than GNPs. Compared with Figure 9 and Appendix A, the impedance matching area of GNPs@Ni_x_S_y_@MoS_2_ composites covers broader frequency and the area is bigger than that of GNPs and GNPs@Ni composites. This shows that Ni_x_S_y_ and MoS_2_ can significantly optimize the impedance matching of GNPs@Ni_x_S_y_@MoS_2_ composites. Compared to Figure 9a,c,e, the order of the impedance matching area is GNPs@Ni_x_S_y_@MoS_2_-2 > GNPs@Ni_x_S_y_@MoS_2_-3 > GNPs@Ni_x_S_y_@MoS_2_-1, which coincides with the corresponding MAP order. For GNPs@Ni_x_S_y_@MoS_2_-1, the minimum RL can be obtained at 6.2 GHz, where α reaches only 100 and *Z* is equal to 1. For GNPs@Ni_x_S_y_@MoS_2_-2, *Z* is close to 1 in the range of 8.5–10.0 GHz with the thickness increase from 1.8 mm to 2.5 mm. The minimum RL value of −43.3 dB can be achieved at 9.1 GHz and 2.2 mm coating thickness, but the α value only reaches 110. While α value reaches a maximum of 750 at 16.0 GHz, *Z* is too small (~0.2). In other words, the minimum RL are not obtained at 16.0 GHz due to the impedance mismatch, which is consistent with Figure 7d. For GNPs@Ni_x_S_y_@MoS_2_-3, the minimum RL values of −39.5 dB and −28.3 dB are achieved at 8.1 GHz (2.4 mm thickness) and 4.8 GHz (4 mm thickness), respectively. Most importantly, the corresponding *Z* values are much close to 1 at 4.8 GHz and 8.1 GHz. Therefore, this clearly infers that impedance mismatching is the limiting factor of GNPs and GNPs@Ni_x_S_y_@MoS_2_ composites’ MAP. In addition, GNPs@Ni_x_S_y_@MoS_2_ composites with different hierarchical structures obtained by sulfurization process have optimized the impedance matching.

In general, there are three kind of pathways (interface polarization, dipoles polarization, and conductive loss) can be contributed to the excellent MAP of GNPs@Ni_x_S_y_@MoS_2_ composites. Firstly, interfacial polarization generates between the interfaces of Ni_x_S_y_-MoS_2_, Ni_x_S_y_-GNPs, and MoS_2_-GNPs promote the EM wave absorption by supercapacitor-like structure and results in fast accumulation of bound charges, due to the high porosities of the hierarchical GNPs@Ni_x_S_y_@MoS_2_ composites. Secondly, several dipoles originate from abundant defects and functional groups of GNPs, during the thermal treatment and hydrothermal reaction process, respectively, that could be responsible for dipole polarization. More important, on the one hand, GNPs are a kind of good conductive, thus, the charges that accumulated on the surface of the interfaces of Ni_x_S_y_-MoS_2_, Ni_x_S_y_-GNPs, and MoS_2_-GNPs can easily transport on the GNPs, which release the interfaces and promote the interface polarization. On the other hand, much EM waves were absorbed by conductive loss that originate from the induced current with the alternating EM wave radiating [61,66]. Therefore, among the multi-interface polarization, dipole polarization, and conductive loss, conductive loss is the most effective attenuation mechanism in GNPs@Ni_x_S_y_@MoS_2_ composites.

## 4. Conclusions

The key limiting factor of MAP of GNPs and GNPs@Ni composites is impedance mismatching owing to high conductivity leading to skin effect, that inhibits the EM wave permeation as revealed in this comparative study. The conductivities were adjusted by simple sulfurization process, resulting in a series of GNPs@Ni_x_S_y_@MoS_2_ composites with different microstructures, that were analyzed via SEM, TEM, XRD, XPS and TGA. With respect to the MAP of GNPs@Ni_x_S_y_@MoS_2_ composites, the optimum RL is −43.3 dB at a thickness of 2.2 mm and the corresponding bandwidth with effective attenuation up to 3.6 GHz. In addition, the absorption mechanism could be responsible for the conductive loss, multi-interface polarization, and dipole polarization. Therefore, this study has not only scientifically revealed the key limiting factor for the inferior MAP of GNPs, but also a new method to optimize impedance matching. In addition, materials with moderate conductivity are still promising to deal with the pollution of EM wave; single component, especially, dielectric materials, are not enough to attenuate EM wave by dielectric loss. Concerning the cost, rational design GNPs-based materials with first-rank component are still a great challenge in the future.

## Figures and Tables

**Figure 1 nanomaterials-09-01403-f001:**
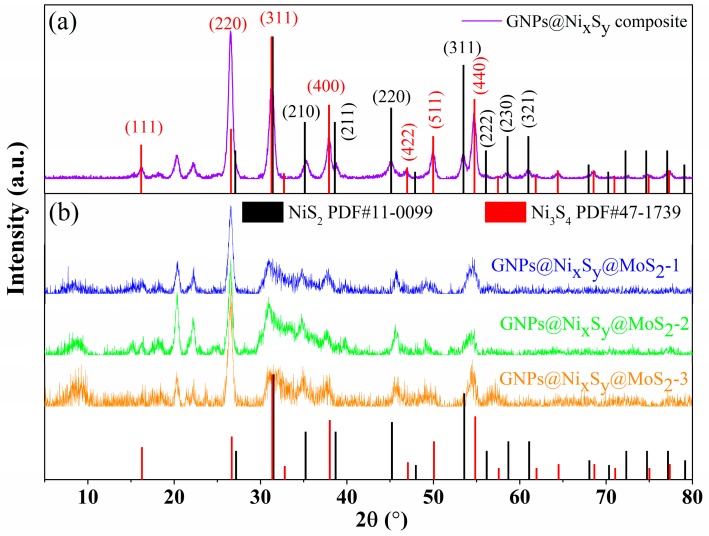
XRD patterns of (**a**) GNPs@Ni_x_S_y_ composite, and (**b**) GNPs@Ni_x_S_y_@MoS_2_ composites.

**Figure 2 nanomaterials-09-01403-f002:**
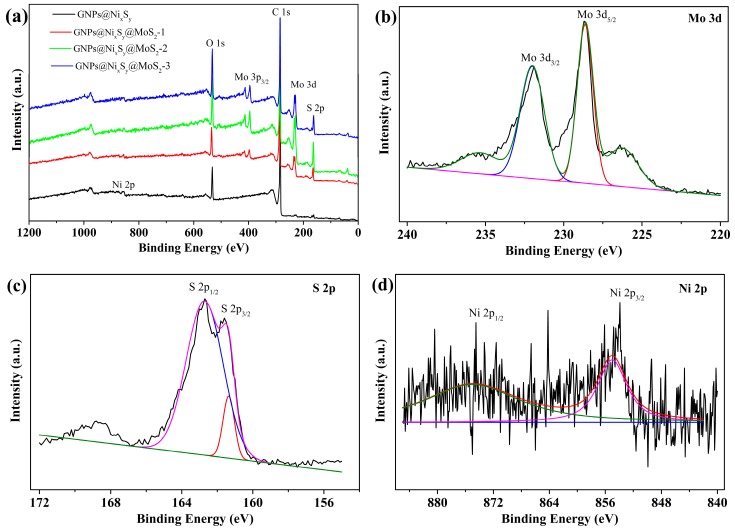
XPS survey spectrum (**a**) and core spectra of Mo 3d (**b**), S 2p (**c**), and Ni 2p (**d**) of GNPs@Ni_x_S_y_@MoS_2_-3.

**Figure 3 nanomaterials-09-01403-f003:**
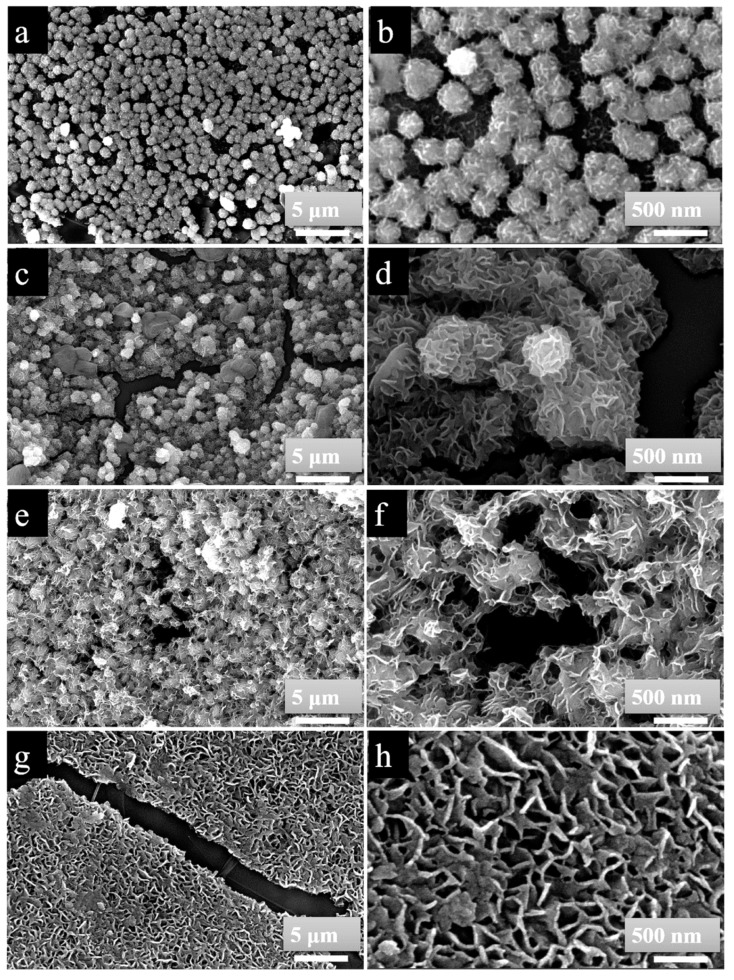
FE-SEM images of GNPs@Ni composite (**a**,**b**), GNPs@Ni_x_S_y_@MoS_2_-1 composite (**c**,**d**), GNPs@Ni_x_S_y_@MoS_2_-2 composite (**e**,**f**), and GNPs@Ni_x_S_y_@MoS_2_-3 composite (**g**,**h**).

**Figure 4 nanomaterials-09-01403-f004:**
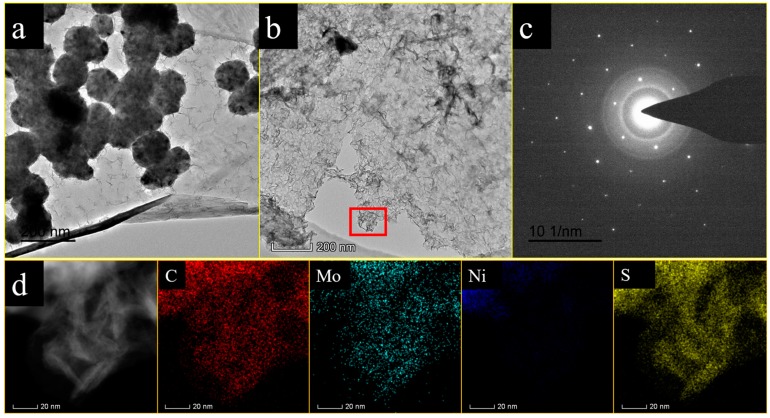
TEM images of (**a**) GNPs@Ni composite and (**b**) GNPs@Ni_x_S_y_@MoS_2_-3 composite. (**c**) Selected area electron diffraction (SAED) pattern of GNPs@Ni_x_S_y_@MoS_2_-3 composite. (**d**) Energy dispersive spectroscopy (EDS) elemental mappings of C, Mo, Ni, and S with corresponding TEM image of GNPs@Ni_x_S_y_@MoS_2_-3.

**Figure 5 nanomaterials-09-01403-f005:**
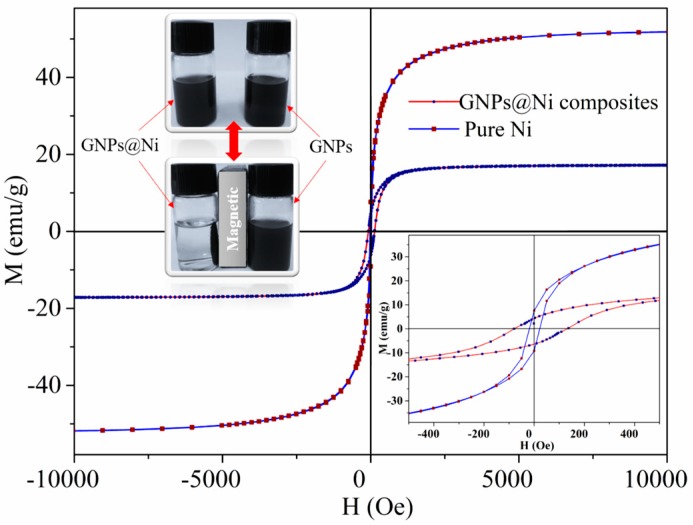
Magnetization hysteresis loops of GNPs@Ni composites and pure Ni nanoparticles.

**Figure 6 nanomaterials-09-01403-f006:**
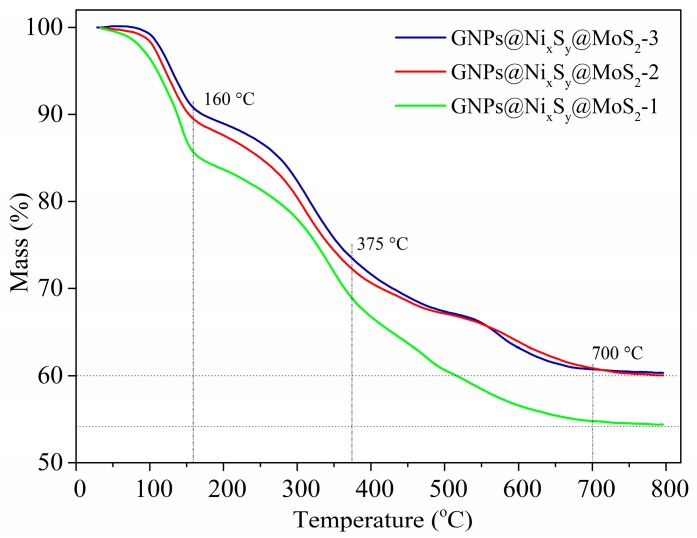
TG curves of as-prepared GNPs@Ni_x_S_y_@MoS_2_ composites in air atmosphere.

**Figure 7 nanomaterials-09-01403-f007:**
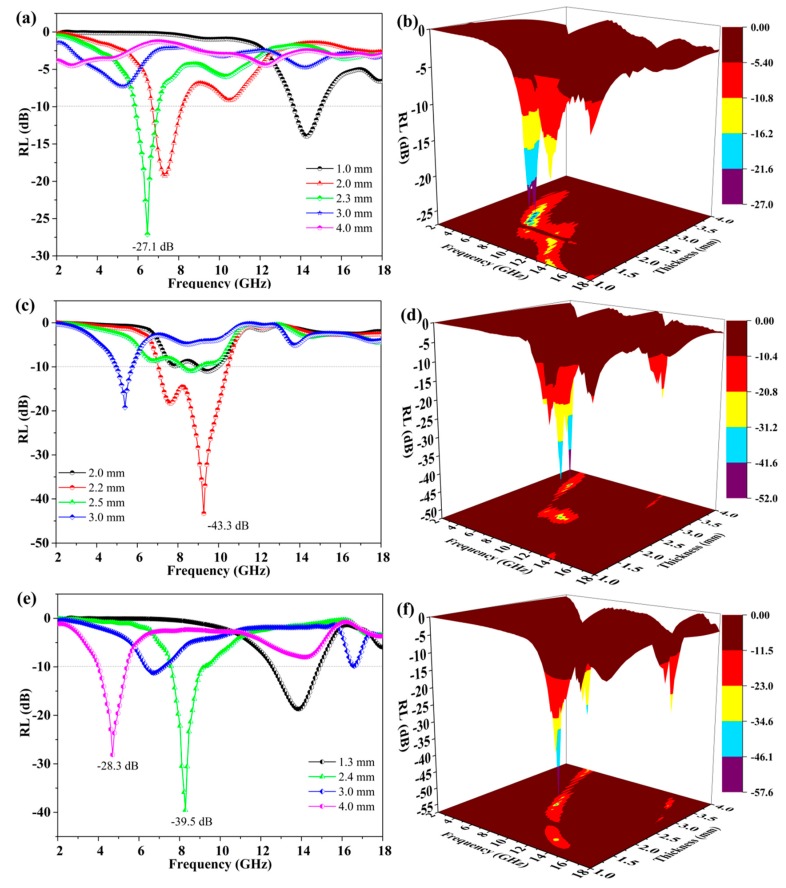
RL and 3D surface plots of GNPs@Ni_x_S_y_@MoS_2_-1 composite (**a**,**b**), GNPs@Ni_x_S_y_@MoS_2_-2 composite (**c**,**d**), GNPs@Ni_x_S_y_@MoS_2_-3 composite (**e**,**f**) at different thicknesses.

**Figure 8 nanomaterials-09-01403-f008:**
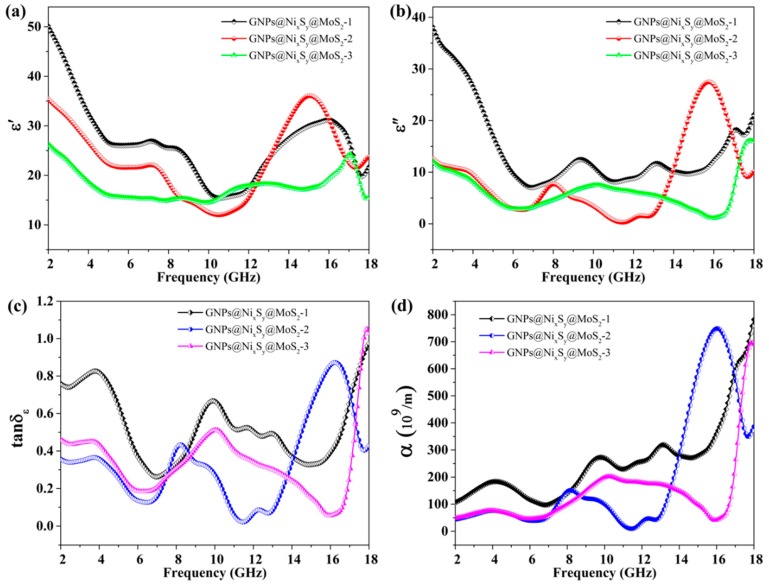
Electromagnetic parameters (**a**,**b**), tangent of dielectric loss (**c**) and α (**d**) of GNPs@Ni_x_S_y_@MoS_2_ composites.

**Figure 9 nanomaterials-09-01403-f009:**
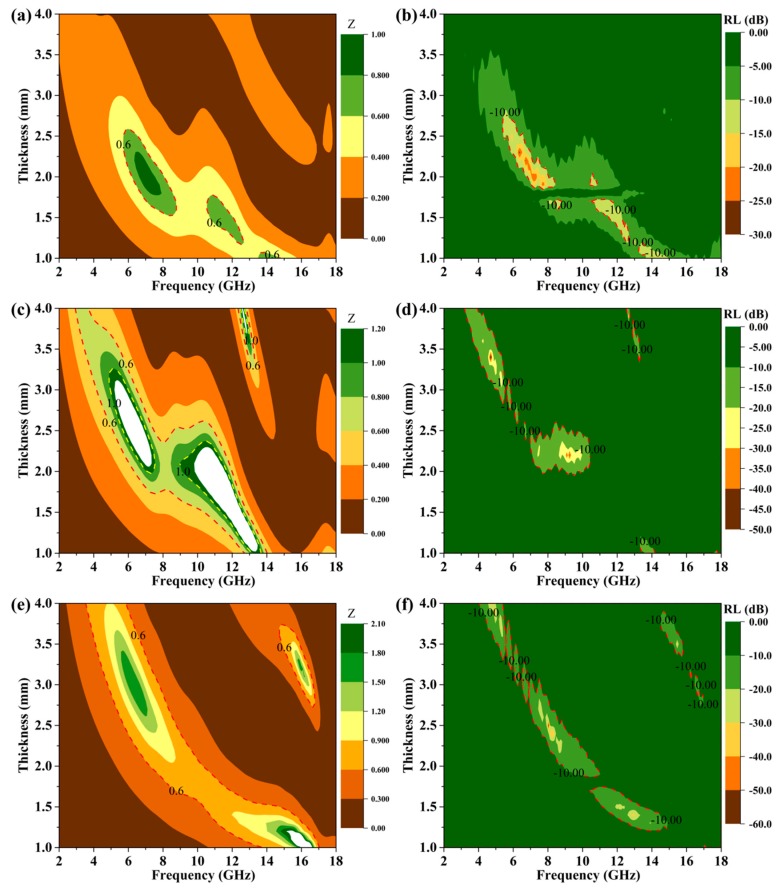
*Z* and RL contour maps of GNPs@Ni_x_S_y_@MoS_2_-1 composite (**a**,**b**), GNPs@Ni_x_S_y_@MoS_2_-2 composite (**c**,**d**), GNPs@Ni_x_S_y_@MoS_2_-3 composite (**e**,**f**).

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
