# Peer review of "Facile Synthesis of GNPs@NixSy@MoS2 Composites with Hierarchical Structures for Microwave Absorption"

_nanomaterials, 2019, doi:10.3390/nano9101403_

Round 1
Reviewer 1 Report
This work collects lots of data on the target subjects, and many exciting aspects are well included. The authors show many characterization data to support the claims. The manuscript is well organized. After the following minor revision, I can accept the publication of this paper. How about material stability? It is better to add more comments on this. The authors show wide-angle XRD patterns for samples. How about the average crystallite sizes? This size is matched with TEM data? Related papers have been reported by different research groups. It is better to cite the following refs to support some related paragraphs in the introduction part. "Considering nano-structural engineering and impedance matching, ..... " It is better to address importance of nano-structural engineering using previous refs. Small 9, 1047, 2013; Small 9, 2520, 2013Langmuir 30, 651, 2014
JACS, 130, 10165, 2008
Journal of Materials Chemistry 16, 3091, 2006
Chemical Communications 47, 6677, 2011
Angew Chem Int Ed, 57, 2018, 8881 Overall the manuscript is well written, but I want to see the authors' perspective on this research in the conclusion part. Some little typo errors are found. Please carefully check the sentences again.
Author Response
Response to Reviewer 1 Comments
Reviewer #1: This work collects lots of data on the target subjects, and many exciting aspects are well included. The authors show many characterization data to support the claims. The manuscript is well organized. After the following minor revision, I can accept the publication of this paper.
Point 1: How about material stability? It is better to add more comments on this 

Response 1: Thank you for the significant question. In accordance with this suggestion, thermogravimetric analysis (TGA) was displayed in Fig. 6. Please check it in the revised muanscript.
Point 2: The authors show wide-angle XRD patterns for samples. How about the average crystallite sizes? This size is matched with TEM data?
Response 2: In this study, we prepared three kinds of GNPs@NixSy@MoS2 composites with different morphologies. The morphologies of as-prepared samples were well illustrated by FE-SEM (as shown in Figure 3). Therefore, the TEM images were obtained from GNPs@NixSy@MoS2-3 by random selection. For GNPs@Ni, the average crystallite sizes of Ni particles obtained from SEM analysis was approximately 150-200 nm, which is matched with TEM data. In any case, the results of SEM are well coincided with the TEM, furthermore, to evaluate the crystallite size, particle size analysis should be considered, actually, NixSy and MoS2 are flower-like particles. Thus, it is very hard to clearly measure the size of as-prepared NixSy and MoS2.
Point 3: Related papers have been reported by different research groups. It is better to cite the following refs to support some related paragraphs in the introduction part. "Considering nano-structural engineering and impedance matching, " It is better to address importance of nano-structural engineering using previous refs. Small 9, 1047, 2013; Small 9, 2520, 2013; Langmuir 30, 651, 2014 JACS, 130, 10165, 2008,Journal of Materials Chemistry 16, 3091, 2006; Chemical Communications 47, 6677, 2011; Angew Chem Int Ed, 57, 2018, 8881
Response 3: Thank you very much for providing us with relevant literatures which are very useful for our research work. These articles have greatly expanded our horizons in the research of thermal conductivity materials, and have given us a lot of new inspiration. We have cited these literatures and compared the results in the article. Detailed contents can be found on page 2 of the article and in references [30-36].
Point 4: Overall the manuscript is well written, but I want to see the authors' perspective on this research in the conclusion part.
Response 4: Thank you for the valuable suggestion. Now, we have revised the conclusion part, please check it in the revised manuscript.
Point 5: Some little typo errors are found. Please carefully check the sentences again.
Response 5: Thank you for the valuable advice. We carefully check the whole article again. And those typo errors were corrected, the corresponding sentences were highlighted in the revised manuscript.

Reviewer 2 Report
Qi et al. present an interesting study of sulfide / graphene-based EM absorbers.Then study should be considered for publication upon coverage of some revision points.
(1) Section 2.2:
- line 78: provide reference on the quoted sensitization,
- line 85: provide concentration of ammonia,
(2) Section 2.3: provide yield of the process.
(3) Figure 1: some peaks below 2 theta of 30 deg. are not assigned - please interpret them.
(4) Section 3.6: for any fits performed, these should be displayed in graphical form along with experimental data. If no fits can be presented, this should be justified.
(5) What is stability of the described materials? Please illustrate with experimental data.
Author Response
Response to Reviewer 2 Comments
Point 1: Section 2.2:
- line 78: provide reference on the quoted sensitization,
- line 85: provide concentration of ammonia,
Response 1: Thank you for your advice. It is indeed necessary to provide reference on the quoted sensitization and the concentration of ammonia. We have added the corresponding reference of sensitization and the concentration of ammonia. For specific text corrections, please see the second page of the article and references [39]
Point 2: Section 2.3: provide yield of the process.
Response 2: Thank you very much for this suggestive advice. According to the experiment record, the weight increasing of GNPs@NixSy composites, GNPs@NixSy@MoS2-1, GNPs@NixSy@MoS2-2 and GNPs@NixSy@MoS2-3 were 20, 170, 175 and 174%, respectively. Thus, based on the simple calculation, we can infer the composition of as-prepared samples were shown in the Table S1.
Point 3: Figure 1: some peaks below 2 theta of 30 deg. are not assigned - please interpret them.
Response 3: Thank you for the significant question. These two peaks located at 2θ = 20.34 (103) and 22.2 (213) are assigned to SiO2 (JCPDS Card no. 79-0430). This may result from that the GNPs@NixSy@MoS2 nanocomposites powders are very fluffy after freeze-drying. When XRD testing was carried out, the fluffy powders might not be compacted enough. Thus, some of the SiO2 diffraction peaks occurred.
Point 4: Section 3.6: for any fits performed, these should be displayed in graphical form along with experimental data. If no fits can be presented, this should be justified.
Response 4: Thank you for the question. The RL curves of samples shown in Fig. 6 were obtained from smoothing performed not from fits performed. These curves were first constructed from the experimental data. Then, we smoothed the curves to keep the Figure looking nicer.
Point 5: What is stability of the described materials? Please illustrate with experimental data.
Response 5: Thank you for the significant question. In accordance with this suggestion, thermogravimetric analysis (TGA) was displayed in Fig. 6. Please check it in the revised manuscript.
